# Sense and Sensitivity Analysis: Simple Post-Hoc Analysis of Bias Due to Unobserved Confounding

Victor Veitch[1] and Anisha Zaveri[2]

[1]*Google Research*
[2]*Weill Cornell Medicine*

## Abstract

It is a truth universally acknowledged that an observed association without known mechanism must be in want of a causal estimate. However, Causal estimates from observational data will be biased in the presence of 'unobserved confounding'. Nevertheless, we might hope that the influence of unobserved confounders is weak relative to a 'large' estimated effect. The purpose of this paper is to develop *Austen plots*, a sensitivity analysis tool to aid such judgments by making it easier to reason about potential bias induced by unobserved confounding. We formalize confounding strength in terms of how strongly the unobserved confounding influences treatment assignment and outcome. For a target level of bias, an Austen plot shows the minimum values of treatment and outcome influence required to induce that level of bias. Austen plots generalize the classic sensitivity analysis approach of Imbens [Imb03]. Critically, Austen plots allow *any* approach for modeling the observed data. We illustrate the tool by assessing biases for several real causal inference problems, using a variety of machine learning approaches for the initial data analysis. Code, demo data, and a tutorial are available at at github.com/anishazaveri/austen_plots.

## 1 Introduction

The high costs of randomized controlled trials coupled with the relative availability of (large scale) observational data motivate attempts to infer causal relationships from observational data. For example, we may wish to use a database of electronic health records to estimate the effect of a treatment. Causal inference from observational data must account for possible *confounders* that influence both treatment assignment and the outcome; e.g., wealth may be a common cause influencing whether a patient takes an expensive drug and whether they recover. Often, causal inference is based on the assumption of 'no unobserved confounding'; i.e., the assumption that the observed covariates include all common causes of the treatment assignment and outcome. This assumption is fundamentally untestable from observed data, but its violation can induce bias in the estimation of the treatment effect—the unobserved confounding may completely or in part explain the observed association. Our aim in this paper is to develop a sensitivity analysis tool to aid in reasoning about potential bias induced by unobserved confounding.

Intuitively, if we estimate a large positive effect then we might expect the real effect is also positive, even in the presence of mild unobserved confounding. For example, consider the association between smoking and lung cancer. One could argue that this association arises from a genetic mutation that predisposes carriers to both an increased desire to smoke and to a greater risk of lung cancer. However, the association between smoking and lung cancer is large—is it plausible that some unknown genetic association could have a strong enough influence to explain the association? Answering such questions requires a domain expert to make a judgment about whether plausible

confounding is "mild" relative to the "large" effect. In particular, the domain expert must translate judgments about the strength of the unobserved confounding into judgments about the bias induced in the estimate of the effect. Accordingly, we must formalize what is meant by strength of unobserved confounding, and to show how to translate judgments about confounding strength into judgments about bias.

A prototypical example, due to Imbens [Imb03] (building on [RR83]), illustrates the broad approach. The observed data consists of a treatment $T$, an outcome $Y$, and covariates $X$ that may causally affect the treatment and outcome. Imbens [Imb03] then posits an additional unobserved binary confounder $U$ for each patient, and supposes that the observed data and unobserved confounder were generated according to:

$$U_i \overset{iid}{\sim} \text{Bern}(1/2)$$

$$T_i \mid X_i, U_i \overset{ind}{\sim} \text{Bern}(\text{sig}(\gamma X_i + \alpha U_i))$$

$$Y_i \mid X_i, T_i, U_i \overset{ind}{\sim} \text{Norm}(\tau T_i + \beta X_i + \delta U_i, \sigma^2).$$

where sig is the sigmoid function. If we had observed $U_i$, we could estimate $(\hat{\tau}, \hat{\gamma}, \hat{\beta}, \hat{\alpha}, \hat{\delta}, \hat{\sigma}^2)$ from the data and report $\hat{\tau}$ as the estimate of the average treatment effect. Since $U_i$ is not observed, it is not possible to identify the parameters from the data. Instead, we make (subjective) judgments about plausible values of $\alpha$—how strongly $U_i$ affects the treatment assignment—and $\delta$—how strongly $U_i$ affects the outcome. Contingent on plausible $\alpha = \alpha^*$ and $\delta = \delta^*$, the other parameters can be estimated. This yields an estimate of the treatment effect $\hat{\tau}(\alpha^*, \delta^*)$ under the presumed values of the sensitivity parameters.

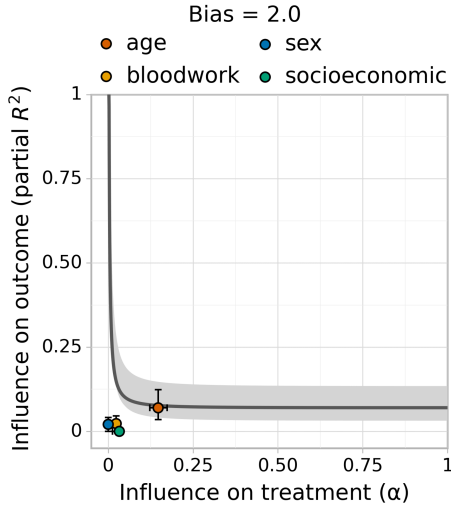

**Figure 1:** Austen plot showing how strong an unobserved confounder would need to be to induce a bias of 2 in an observational study of the effect of combination blood pressure medications on diastolic blood pressure [Dor+16]. We chose this bias to equal the nominal average treatment effect estimated from the data. We model the outcome with Bayesian Additive Regression Trees and the treatment assignment with logistic regression—Austen plots accommodate any choice of models. The curve shows all values treatment and outcome influence that would induce a bias of 2. The colored dots show the influence strength of (groups of) observed covariates, given all other covariates. For example, an unobserved confounder with as much influence as the patient's age might induce a bias of about 2.

The approach just outlined has a major drawback: it relies on a parametric model for the full data generating process. The assumed model is equivalent to assuming that, had $U$ been observed, it would have been appropriate to use logistic regression to model treatment assignment, and linear regression to model the outcome. This assumption also implies a simple, parametric model for the relationships governing the observed data. This restriction is out of step with modern practice, where we use flexible machine-learning methods to model these relationships. For example, the assumption forbids the use of neural networks or random forests, though such methods are often state-of-the-art for causal effect estimation.

**Austen plots** The purpose of this paper is to introduce *Austen plots*, an adaptation of Imbens' approach that fully decouples sensitivity analysis and modeling of the observed data. An example Austen plot is shown in Figure 1. The high-level idea is to posit a generative model that uses a simple, interpretable parametric form for the influence of the unobserved confounder, but that *puts no constraints on the model for the observed data*. We then use the parametric part of the model to formalize "confounding strength" and to compute the induced bias as a function of the confounding.

We further adapt two innovations pioneered by Imbens [Imb03]. First, we find a parameterization of the model so that the sensitivity parameters, measuring strength of confounding, are on a standardized, unitless scale. This allows us to compare the strength of hypothetical unobserved confounding to the strength of observed covariates, measured from data. Second, we plot the curve of all values of the sensitivity parameter that would yield given level of bias. This moves the analyst judgment from "what are plausible values of the sensitivity parameters?" to "are sensitivity parameters this extreme plausible?"

Figure 1, an Austen plot for an observational study of the effect of combination medications on diastolic blood pressure, illustrates the idea. A bias of 2 would suffice to undermine the qualitative conclusion that the blood-pressure treatment is effective. Examining the plot, an unobserved confounder as strong as age could induce this amount of confounding, but no other (group of) observed confounders has so much influence. Accordingly, if a domain expert thinks an unobserved confounder as strong as age is unlikely then they may conclude that the treatment is likely effective. Or, if such a confounder is plausible, they may conclude that the study fails to establish efficacy.

The purpose of this paper is adapting Imbens' sensitivity analysis approach to allow for arbitrary models for observed data. The contributions are: 1. Positing a generative model that is both easily interpretable and where the required bias calculations are tractable. 2. Deriving a reparameterization that standardizes the scale of influence strength, and showing how to estimate the influence strength of observed covariates for reference. And, 3. illustrative examples showing that Austen plots preserve the key elements of Imbens' approach and are informative about sensitivity to unobserved confounding in real-world data.

The key advantages of Austen plots as a sensitivity analysis method are[1] 1. Plausibility judgments are made on directly interpretable quantities, the total confounding influence on $Y$ and $T$. Additionally, the Austen plot model does not rely on the detailed nature of the unobserved confounding— there may be one or many unobserved confounders, with any sort of distribution—all that matters is the total confounding influence. 2. The unobserved strength of confounding can be directly compared to the strength of observed covariates. 3. The method is entirely post-hoc. That is, the analyst does not need to consider any aspect of the sensitivity analysis when modeling the observed data. In particular, producing Austen plots requires *only predictions* from the data models. We provide software and a tutorial for producing the plots.[2]

**Notation** For concreteness, we focus on the estimation of the average effect of a binary treatment. The data are generated independently and identically $(Y_i, T_i, X_i, U_i) \overset{\text{iid}}{\sim} P$, where $U_i$ is not observed and $P$ is some unknown probability distribution. The average treatment affect (ATE) is

$$\text{ATE} = \mathbb{E}[Y \mid \text{do}(T = 1)] - \mathbb{E}[Y \mid \text{do}(T = 0)].$$

The use of Pearl's do notation indicates that the effect of interest is causal. The results that follow can also be simply adapted to the average treatment effect on the treated, see appendix A.

The traditional approach to causal estimation assumes that the observed covariates $X$ contain all common causes of $Y$ and $T$. If this 'no unobserved confounding' assumption holds, then the ATE is equal to parameter, $\tau$, of the observed data distribution, where

$$\tau = \mathbb{E}[\mathbb{E}[Y \mid X, T = 1] - \mathbb{E}[Y \mid X, T = 0]]. \tag{1.1}$$

The parameter $\tau$ can be estimated from a finite data sample. The general approach proceeds in two steps. First, we produce estimates $\hat{g}$ and $\hat{Q}$ for the propensity score $g$ and the conditional expected outcome $Q$, where

**Definition 1.** The *propensity score* $g$ is $g(x) = \text{P}(T = 1 \mid X = x)$ and the *conditional expected outcome* $Q$ is $Q(t, x) = \mathbb{E}[Y \mid T = t, X = x]$.

In modern practice, $Q$ and $g$ are often estimated by fitting flexible machine learning models. The second step is to plug the estimated $\hat{Q}$ and $\hat{g}$ in to some downstream estimator $\hat{\tau}$. For example, following 1.1, the estimator

$$\hat{\tau}^Q = \frac{1}{n} \sum_i \hat{Q}(1, x_i) - \hat{Q}(0, x_i),$$

is a natural choice. Other estimators incorporate $\hat{g}$.

We are interested in the case of possible unobserved confounding. That is, where $U$ causally affects $Y$ and $T$. If there is unobserved confounding then the parameter $\tau$ is not equal to the ATE, so $\hat{\tau}$ is a biased estimate. Inference about the ATE then divides into two tasks. First, the statistical task:

estimating $\tau$ as accurately as possible from the observed data. And, second, the causal (domain-specific) problem of assessing $\mathtt{bias} = \mathtt{ATE} - \tau$. We emphasize that our focus here is bias due to causal misidentification, not the statistical bias of the estimator. Our aim is to reason about the bias induced by unobserved confounding—the second task—in a way that imposes no constraints on the modeling choices for $\hat{Q}$, $\hat{g}$ and $\hat{\tau}$ used in the initial analysis.

## 2 Sensitivity Model

Our sensitivity analysis should impose no constraints on how the *observed* data is modeled. However, sensitivity analysis demands some assumption on the relationship between the observed data and the *unobserved* confounder. It is convenient to formalize such assumptions by specifying a probabilistic model for how the data is generated. The strength of confounding is then formalized in terms of the parameters of the model (the sensitivity parameters). Then, the bias induced by the confounding can be derived from the assumed model. Our task is to posit a generative model that both yields a useful and easily interpretable sensitivity analysis, and that avoids imposing any assumptions about the observed data.

To begin, consider the functional form of the sensitivity model used by Imbens [Imb03].

$$\operatorname{logit} P(T = 1 \mid x, u) = h(x) + \alpha u \tag{2.1}$$
$$\mathbb{E}[Y \mid t, x, u] = l(t, x) + \delta u, \tag{2.2}$$

for some functions $h$ and $l$. That is, the propensity score is logit-linear in the unobserved confounder, and the conditional expected outcome is linear.

By rearranging (2.1) to solve for $u$ and plugging in to (2.2), we see that it's equivalent to assume $\mathbb{E}[Y \mid t, x, u] = \tilde{l}(t, x) + \tilde{\delta} \operatorname{logit} P(T = 1 \mid x, u)$. That is, the unobserved confounder $u$ only influences the outcome through the propensity score. Accordingly, by positing a distribution on $P(T = 1 \mid x, u)$ directly, we can circumvent the need to explicitly articulate $U$ (and $h$).

**Definition 2.** Let $\tilde{g}(x, u) = P(T = 1 \mid x, u)$ denote the propensity score given observed covariates $x$ and the unobserved confounder $u$.

The insight is that we can posit a sensitivity model by defining a distribution on $\tilde{g}$ directly. The logit-linear model does not directly lead to a tractable sensitivity analysis. Instead, we choose:

$$\tilde{g}(X, U) \mid X \sim \operatorname{Beta}(g(X)(1/\alpha - 1), (1 - g(X))(1/\alpha - 1)).$$

The sensitivity parameter $\alpha$ plays the same role as in Imbens' model: it controls the influence of the unobserved confounder $U$ on treatment assignment. When $\alpha$ is close to 0 then $\tilde{g}(X, U) \mid X$ is tightly concentrated around $g(X)$, and the unobserved confounder has little influence. That is, $U$ minimally affects our belief about who is likely to receive treatment. Conversely, when $\alpha$ is close to 1 then $\tilde{g}$ concentrates near 0 and 1; i.e., knowing $U$ would let us accurately predict treatment assignment. Indeed, it can be shown that $\alpha$ is the change in our belief about how likely a unit was to have gotten the treatment, given that they were actually observed to be treated (or not):

$$\alpha = \mathbb{E}[\tilde{g}(X, U) \mid T = 1] - \mathbb{E}[\tilde{g}(X, U) \mid T = 0]. \tag{2.3}$$

With the $\tilde{g}$ model in hand, we define our sensitivity model:

**Assumption 1** (Sensitivity Model)**.**

$$\tilde{g}(X, U) \mid X \sim \operatorname{Beta}(g(X)(1/\alpha - 1), (1 - g(X))(1/\alpha - 1))$$
$$T \mid X, U \sim \operatorname{Bern}(\tilde{g}(X, U))$$
$$\mathbb{E}[Y \mid T, X, U] = Q(T, X) + \delta\big(\operatorname{logit} \tilde{g}(X, U) - \mathbb{E}[\operatorname{logit} \tilde{g}(X, U) \mid X, T]\big).$$

This model has been constructed to satisfy the requirement that the propensity score and conditional expected outcome are the $g$ and $Q$ actually present in the observed data:

$$P(T = 1 \mid X) = \mathbb{E}[\mathbb{E}[T \mid X, U] \mid X] = \mathbb{E}[\tilde{g}(X, U) \mid X] = g(X)$$
$$\mathbb{E}[Y \mid T, X] = \mathbb{E}[\mathbb{E}[Y \mid T, X, U] \mid T, X] = Q(T, X).$$

The sensitivity parameters are $\alpha$, controlling the dependence between the unobserved confounder the treatment assignment, and $\delta$, controlling the relationship with the outcome. In effect, by making an assumption about the propensity score directly, we have sidestepped the need to explicitly articulate the parts of the observed/unobserved relationship that are not actually relevant for the treatment effect estimation.

**Bias** We now turn to calculating the bias induced by unobserved confounding. By assumption, $X$ and $U$ together suffice to render the average treatment effect identifiable as:

$$\mathtt{ATE} = \mathbb{E}[\mathbb{E}[Y \mid T = 1, X, U] - \mathbb{E}[Y \mid T = 0, X, U]].$$

Plugging in our sensitivity model yields,

$$\mathtt{ATE} = \mathbb{E}[Q(1, X) - Q(0, X)] + \delta(\mathbb{E}[\text{logit } \tilde{g}(X, U) \mid X, T = 1] - \mathbb{E}[\text{logit } \tilde{g}(X, U) \mid X, T = 0]).$$

The first term is the observed-data estimate $\tau$, so

$$\mathtt{bias} = \delta(\mathbb{E}[\text{logit } \tilde{g}(X, U) \mid X, T = 1] - \mathbb{E}[\text{logit } \tilde{g}(X, U) \mid X, T = 0]).$$

Then, by invoking Beta-Bernoulli conjugacy and standard Beta identities, we arrive at,

**Theorem 3.** *Under our sensitivity model, Assumption 1, an unobserved confounder with influence $\alpha$ and $\delta$ induces bias in the estimated treatment effect equal to*

$$\mathtt{bias} = \delta\mathbb{E}\big[\psi\big(g(X)(1/\alpha - 1) + 1\big) - \psi\big((1 - g(X))(1/\alpha - 1)\big)$$
$$- \psi\big(g(X)(1/\alpha - 1)\big) + \psi\big((1 - g(X))(1/\alpha - 1) + 1\big)\big],$$

*where $\psi$ is the digamma function*

**Reparameterization** The model in the previous section provides a formalization of confounding strength and tells us how much bias is induced by a given strength of confounding. This lets us translate judgments about confounding strength to judgments about bias. However, $\delta$ may be difficult to interpret. Following Imbens [Imb03], we will reexpress the outcome-confounder strength in terms of the partial coefficient of determination:

$$R^2_{Y,\text{par}}(\alpha, \delta) = \frac{\mathbb{E}(Y - Q(T, X))^2 - \mathbb{E}(Y - \mathbb{E}[Y \mid T, X, U])^2}{\mathbb{E}(Y - Q(T, X))^2}.$$

This parameterization has two advantages over $\delta$. First, $R^2_{Y,\text{par}}$ has a familiar interpretation—the proportion of previously unexplained variation in $Y$ that is explained by the unobserved covariate $U$. Second, $R^2_{Y,\text{par}}$ has a fixed, unitless scale—enabling easy comparisons with reference values.

The key to computing the reparameterization is the following result (proof in appendix):

**Theorem 4.** *Under our sensitivity model, Assumption 1, the outcome influence is*

$$R^2_{Y,\text{par}}(\alpha, \delta) = \delta^2 \sum_{t=0}^{1} \frac{\mathbb{E}\big[\psi_1\big(g(X)^t(1 - g(X))^{1-t}(1/\alpha - 1) + 1[T = t]\big)\big]}{\mathbb{E}[(Y - Q(T, X))^2]},$$

*where $\psi_1$ is the trigamma function.*

We do not reparameterize the strength of confounding on treatment assignment because, by design, $\alpha$ is already interpretable and on a fixed, unitless scale.

**Estimating bias** In combination, Theorems 3 and 4 yield an expression for the bias in terms of $\alpha$ and $R^2_{Y,\text{par}}$. In practice, we can estimate the bias induced by confounding by fitting models for $\hat{Q}$ and $\hat{g}$ and replacing the expectations by means over the data. To avoid problems associated with overfitting, we recommend a data splitting approach. Namely, split the data into $k$ folds and, for each fold, estimate $Q(t_i, x_i)$ and $g(x_i)$ by fitting the $\hat{Q}$ and $\hat{g}$ models on the other $k - 1$ folds.

## 3 Calibration using observed data

The analyst must make judgments about the influence a hypothetical unobserved confounder might have on treatment assignment and outcome. To calibrate such judgments, we'd like to have a reference point for how much the observed covariates influence the treatment assignment and outcome.

In the sensitivity model, the degree of influence is measured by partial $R_Y^2$ and $\alpha$. We want to measure the degree of influence of an observed covariate $Z$ given the other observed covariates $X \backslash Z$.

For the outcome, this can be measured as:

$$R_{Y \cdot Z | T, X \backslash Z}^2 := \frac{\mathbb{E}(Y - \mathbb{E}[Y \mid T, X \backslash Z])^2 - \mathbb{E}(Y - Q(T, X))^2}{\mathbb{E}(Y - \mathbb{E}[Y \mid T, X \backslash Z])^2}.$$

In practice, estimate the quantity by fitting a new regression model $\hat{Q}_Z$ that predicts $Y$ from $T$ and $X \backslash Z$. Then we compute

$$R_{Y \cdot Z | T, X \backslash Z}^2 = \frac{\frac{1}{n} \sum_i (y_i - \hat{Q}_Z(t_i, x_i \backslash z_i))^2 - \frac{1}{n} \sum_i (y_i - \hat{Q}(t_i, x_i))^2}{\frac{1}{n} \sum_i (y_i - \hat{Q}_Z(t_i, x_i \backslash z_i))^2}.$$

It is less clear how to produce the analogous estimate for the influence on treatment assignment. To facilitate the estimation, we reexpress $\alpha$ in a more convenient form (proof in appendix):

**Theorem 5.** *Under our sensitivity model, Assumption 1,*

$$\alpha = 1 - \frac{\mathbb{E}[\tilde{g}(X, U)(1 - \tilde{g}(X, U))]}{\mathbb{E}[g(X)(1 - g(X))]}.$$

Then, we can measure influence of observed covariate $Z$ on treatment assignment given $X \backslash Z$ in an analogous fashion to the outcome. We define $g_{X \backslash Z}(X \backslash Z) = P(T = 1 \mid X \backslash Z)$, then fit a model for $g_{X \backslash Z}$ by predicting $T$ from $X \backslash Z$, and estimate

$$\hat{\alpha}_{Z | X \backslash Z} = 1 - \frac{\frac{1}{n} \sum_i \hat{g}(x_i)(1 - \hat{g}(x_i))}{\frac{1}{n} \sum_i \hat{g}_{X \backslash Z}(x_i \backslash z_i)(1 - \hat{g}_{X \backslash Z}(x_i \backslash z_i))}.$$

**Grouping covariates** The estimated values $\hat{\alpha}_{X \backslash Z}$ and $\hat{R}_{Y, X \backslash Z}^2$ measure the influence of $Z$ conditioned on all the other confounders. In some cases, this can be misleading. For example, if some piece of information is important but there are multiple covariates providing redundant measurements, then the estimated influence of each covariate will be small. To avoid this, we suggest grouping together related or strongly dependent covariates and computing the influence of the entire group in aggregate. For example, grouping income, location, and race as 'socioeconomic variables'.

## 4   Examples

We now examine several examples of Austen plots for sensitivity analysis, showing: (1) We preserve the qualitative usefulness of Imbens' approach, without any modeling restrictions. (2) Austen plots are informative about bias due to unobserved confounding in real observational studies. (3) The bias estimates tend to be conservative.

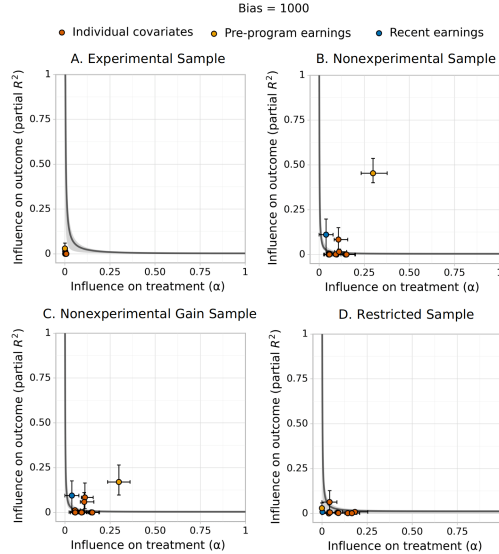

**Figure 2:** Austen plots preserve the qualitative conclusions of Imbens' analysis without imposing any restriction on the modeling of the observed data. In each plot, the black solid line indicates the partial $R^2$ and $\alpha$ values that would induce a bias of at least \$1000. Each plot also includes estimates for the strength of confounding for each of the nine covariates (red circles) as well as recent lag in earnings (RE75 and pos75, yellow circles), and the all preprogram earnings (RE74, pos74, RE75, pos75, green circles).

**Imbens' analysis** To demonstrate the use of Austen plots, we replicate Imbens [Imb03] example and produce sensitivity plots for variations on the LaLonde job training data [LaL86]. We use exactly the same data splitting and adjustment sets as Imbens [Imb03]. We find that the conclusions about the effects of unobserved confounding are substantively the same as Imbens [Imb03]. That is, we arrive at sensible sensitivity conclusions while liberating ourselves from the need for parametric assumptions on the observed data. We report bias for the average treatment effect on the treated.

The original purpose of the LaLonde job training data was to analyze the effect of a job training program on the annual earnings of a participant. The data consists of both an experimental (randomly assigned) part, and an observational sample from the Panel Study of Income Dynamics (PSID). We test on (1) the experimental sample, (2) the experimental treated with observational controls, (3) the same as 2, except with outcome defined as change in earnings since 1974, and (4) the same as 2, except individuals with high earnings pretreatment (>$5000) are dropped. We adjust for: married, age, education, race, and earnings in 1974 and 1975. There are large differences in these background covariates between the experimental sample and the PSID controls—this is a main challenge for the LaLonde setup.

Deviating from Imbens, we fit random forests for $\hat{Q}$ and $\hat{g}$. This demonstrates the sensitivity analysis in the case where the observed data model does not have a simple parametric specification.

Austen plots for these analyses are displayed in Figure 2. Following Imbens, we choose a bias of $1000 (for context, the effect estimate from the RCT is about $1800). The experimental sample (panel A) is robust to unobserved confounding: inducing a bias of $1000 would require an unobserved confounder with a much stronger effect than any of the measured covariates or earning variables. By contrast, the non-experimental samples (panels B and C) are much more sensitive to unobserved confounding. Several of the covariates, if unobserved, would suffice to bias the estimate by $1000. Note that the sensitivity curves are the same for both B and C, since the outcome is just a linear transformation. Finally, the restricted sample (panel D) is both significantly more robust to bias than the full non-experimental samples, and the influence of the observed covariates is much reduced. Imposing the restriction mitigates the treatment-control population mismatch.

**Practical relevance** Figure 3 shows Austen plots for two effects estimated from observational data. The first study is based on data from the Infant Health and Development Program (IHDP) [BG+92], an experiment targeted at low-birth-weight, premature infants that provided child care and home visits. We look at a study measuring the effect of the level of participation in IHDP child development centers in the two years preceding an IQ test on the outcome of the IQ test [Hil11, §6.1]. Level of participation is not randomly assigned, so Hill [Hil11] estimates the effect by using Bayesian Additive Regression Trees (BART) [Chi+10] to control for a range of covariates.

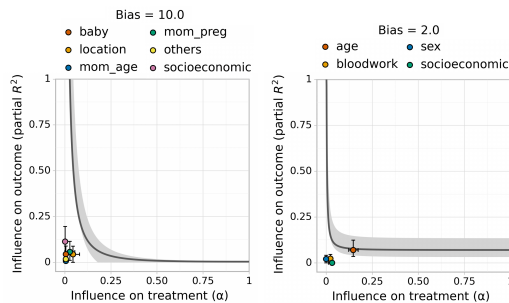

The second plot corresponds to the estimate of the effect of combination blood pressure medications on diastolic blood pressure described in [Dor+16]. The data is derived from an American survey that includes a variety of socioeconomic and health factors. We again use BART.

The Austen plots are informative for these examples. In the first case, the Austen plot increases our confidence in the qualitative result. In the second

**Figure 3:** Austen plots are informative when applied to real data analysis. The left-hand plot is for the estimated effect of IHDP participation level on child IQ. The conclusions of this study seem robust to unobserved confounding—even the observed covariate groups do not have sufficient influence to undo the qualitative conclusion of the model. The right-hand plot is for the estimated effect of combination treatment on diastolic blood pressure. In this case, whether the study conclusions are reliable depends on whether an unobserved confounder as influential as age is credible—we should consult with an expert. In both cases, we model the outcome with Bayesian Additive Regression Trees, and the propensity score with logistic regression.

case, it suggests we should be cautious about the conclusions unless unobserved confounders as strong as age are deemed unlikely.

**Table 1:** The sensitivity model tends to be conservative in its bias estimates. Bias estimates for leaving out a confounding covariate are computed according to the sensitivity model (using the left-out covariate data) and by comparing non-parametric effect estimates from the full data ($\tau_x$), and the left-out covariate data ($\tau_{x \setminus z}$). In all cases, the sensitivity model estimate is larger.

| | Study: Omitted covariate: | LaLonde Restricted Education | Blood Pressure Age | IHDP Socioeconomic |
|---|---|---|---|---|
| $\tau_x$ | | 2508.63 | $-2.33$ | 12.72 |
| $\tau_{x \setminus z}$ | | 1982.54 | $-2.86$ | 13.35 |
| Nonparametric `bias` | | 526.09 | 0.53 | $-0.63$ |
| Sensitivity Model \|`bias`\| | | 986.90 | 1.91 | 0.75 |

**Sensitivity model conservatism** Any sensitivity analysis must be predicated on some assumption about the influence of the unobserved confounder. The bias curves and influence estimates in Austen plots are contingent on the assumed sensitivity model, Assumption 1. We motivated our particular choice by simplicity and tractibility. We also expect that our associated sensitivity model will often yield conservative values for bias; i.e., the bias anticipated by the sensitivity model is higher than the true bias induced by the real, physical, mechanism of confounding. The reason is that bias is monotonically increasing in both treatment and outcome influence. In reality, hidden confounders can have more complicated relationships that 'cancel out'. For example, the effect of age in the blood pressure example might be: blood pressure increases with age, but young patients don't take their medication (preferring diet and exercise), middle age patients take it at a base rate, and old patients don't take the medication (fatalism). These effects cancel out somewhat, reducing the bias induced by failing to adjust for age. Assumption 1 does not allow for such cancellations.

To test conservativism, we create deliberately confounded datasets by removing an observed confounder from our baseline data. We compute the bias anticipated by our model, $\texttt{bias}(R^2_{Y,X \setminus Z}, \alpha_{X \setminus Z})$, using the measured influence strength of the covariate. We compute a non-parametric estimate of the bias by estimating the effect with the full data, estimating the effect with the deliberately confounded data, and taking the difference. The results are shown in table 1, and confirm the conservatism-in-practice. This increases our confidence that when an Austen plot suggests robustness to unobserved confounding we do indeed have such robustness.

## 5 Related Work

We survey a few approaches to sensitivity analysis; for reviews, see [Liu+13; Ric+14].

Austen plots build on sensitivity analysis based on parametric models in the style of Imbens [Imb03]. These typically assume some relatively simple parametric latent variable model, where the latent variable is the unobserved confounder. Dorie et al. [Dor+16] extends an Imbens-like approach to accomodate BART as the outcome model. Cinelli and Hazlett [CH20] allow for arbitrary kinds of confounders and propensity score models, but require that the outcome is modeled with linear regression. Cinelli et al. [Cin+19] assume a causal DAG where all relationships are linear. They make further assumptions about the DAG and use identification tools to derive effect (hence, bias) estimates from the assumptions.

An alternative is to relax parametric assumptions at the price of requiring the analyst to make judgments about more abstract sensitivity parameters [e.g., Fra+19; She+11; HS13; BK19]. Franks et al. [Fra+19] allow arbitrary models for the initial analysis. Their sensitivity model is adapted from the missing data literature, and requires the analyst to specify $P(T = t \mid Y(1-t), X)$—the posterior belief about probability of treatment assignment had the counterfactual outcome under no-treatment been observed. Shen et al. [She+11] give an inverse-propensity-weighted specific approach where the sensitivity parameters are $\text{var} \tilde{g}/g$ and the correlation between $\tilde{g}/g$ and the counterfactual outcomes. Bonvini and Kennedy [BK19] assume each unit is confounded or not, and take the proportion of confounding as the sensitivity parameter.

Another approach estimates the range of effects consistent with a bound on confounding [e.g., Ros10; Zha+19; Yad+18]. For instance, Rosenbaum [Ros10] and Yadlowsky et al. [Yad+18] assume $\tilde{g}$ is logit-linear in an unobserved confounder $U$ with $U \in_R [-1, 1]$ and derive effect bounds

from assumed bounds on the coefficient of $U$. This avoids parametric assumptions on observed data and on the relationship between $U$ and $Y$. However, specifying the coefficient a priori can be challenging. This approach is analogous to considering only partial $R^2$ equal 1 point on the bias curve.

There is also substantial work on calibrating sensitivity analyses using observed data [e.g., Fra+19; HS13; CH20], including some concerns raised in [CH20, §6.2] (discussed below).

# 6    Limitations and Future Directions

We have developed Austen plots as a tool for assessing sensitivity to unobserved confounding. Austen plots are most useful in situations where the conclusion from the plot would be 'obvious' to a domain expert. For instance, the LaLonde RCT plot shows that a confounder would have to be much stronger than the observed covariates to induce substantial bias. Similarly, the LaLonde observational plot shows that confounders similar to the observed covariates could induce substantial bias. Such conclusions would not be affected by mild perturbations of the dots or the line. By contrast, Austen plots should not be used to draw conclusions such as, "I think a latent confounder could only be 90% as strong as 'age', so there is evidence of a small non-zero effect". Such nuanced conclusions might depend on issues such as the particular sensitivity model we use, or finite-sample variation of our bias and influence estimates, or on incautious interpretation of the calibration dots. Drawing precise quantitative (rather than qualitative) conclusions about induced bias from Austen plots would require careful consideration of these issues, and expert statistical guidance. Hence, Austen plots should be used mainly with domain experts to guide qualitative conclusions ("this job program likely works", "this study doesn't establish drug efficacy").

**Inference**    In practice, producing Austen plots requires estimating the parameters of the curves and dots using a finite data sample. However, the plots may be misleading if there is significant finite-sample estimation error. In this paper, we use a simple plug-in estimator that substitutes $\hat{Q}, \hat{g}$ for the true $Q$ and $g$. There are two main limitations: the simple estimator may be statistically inefficient, and uncertainty quantification is difficult. The plots in this paper used bootstrap confidence intervals (100 samples). This is computationally burdensome for the machine-learning models that motivate the paper. Further, the bootstrapping for the influence statistics is finnicky—e.g., care must be taken to group replicated samples together in the $k$-folding. It would be very useful to develop (non-parametric) efficient estimators for the key quantities that also admit explicit variance estimators.

**Calibration using observed data**    The interpretation of the observed-data calibration requires some care. The sensitivity analysis requires the analyst to make judgements about the strength of influence of the unobserved confounder $U$, *conditional on the observed covariates* $T, X$. However, we report the strength of influence of observed covariate(s) $Z$, *conditional on the other observed covariates* $T, X \backslash Z$. The difference in conditioning sets can have subtle effects.

Cinelli and Hazlett [CH20] give an example where $Z$ and $U$ are identical variables in the true model, but where influence of $U$ given $T, X$ is larger than the influence of $Z$ given $T, X \backslash Z$. (The influence of $Z$ given $T, X \backslash Z, U$ would be the same as the influence of $U$ given $T, X$). Accordingly, an analyst is *not* justified in a judgement such as, "I know that $U$ and $Z$ are very similar. I see $Z$ has substantial influence, but the dot is below the line. Thus, $U$ will not undo the study conclusions." In essence, if the domain expert suspects a strong interaction between $U$ and $Z$ then naively eyeballing the dot-vs-line position may be misleading. A particular subtle case is when $U$ and $Z$ are independent variables that both strongly influence $T$ and $Y$. The joint influence on $T$ creates an interaction effect between them when $T$ is conditioned on (the treatment is a collider). This affects the interpretation of $R^2$. Indeed, we should generally be skeptical of sensitivity analysis interpretation when it is expected that a strong confounder has been omitted. In such cases, our conclusions may depend substantively on the particular form of our sensitivity model, or other unjustifiable assumptions.

Although the interaction problem is conceptually important, its practical significance is unclear. We often expect the opposite effect: if $U$ and $Z$ are dependent (e.g., race and wealth) then omitting $U$ should increase the apparent importance of $Z$—leading to a conservative judgement. It would be useful for future work to articulate specific situations where Austen plots cannot be trusted, beyond "we expect a strong $U$ is missing". As part of this, it could be useful to derive a bias formula in terms of $R^2_{Y \cdot U | X}$ instead of $R^2_{Y \cdot U | X, T}$. This would eliminate the need to consider the collider naunce.

## 7  Societal Consequences

This paper addressed sensitivity analysis for causal inference. We have extended Imbens' approach to allow the use of arbitrary machine-learning methods for the data modeling. Austen plots provide an entirely post-hoc and blackbox manner of conducting sensitivity analysis. In particular, they make it substantially simpler to perform sensitivity analysis. This is because the initial analysis can be performed without have a sensitivty analysis already in mind, and because producing the sensitivity plots only requires predictions from models that the practitioner has fit anyways.

The ideal positive consequence is that routine use of Austen plots will improve the credibility of machine-learning based causal inferences from observational data. Austen plots allow us to both use state-of-the-art models for the observed part of the data, and to reason coherently about the causal effects of potential unobserved confounders. The availability of such a tool may speed the adoption of machine-learning based causal inference for important real-world applications (where, so far, adoption has been slow).

On the negative side, an accelerated adoption of machine-learning methods into causal practice may be undesirable. This is simply because the standards of evidence and evaluation used in common machine-learning practice do not fully reflect the needs of causal practice. Austen plots partially bridge this gap, but they just one of the elements required to establish credibility.

## 8  Funding

Victor Veitch was supported in part by the Government of Canada through an NSERC PDF.

## Footnotes

[1]See section 5 for a more detailed comparison with related work.

[2]github.com/anishazaveri/austen_plots

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
