[Supplementary Material]

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

There are a wealth of approaches to sensitivity analysis. The most closely related approaches to ours are sensitivity analysis based on parametric models in the style of Imbens [Imb03]. These typically assume some relatively simple parametric latent variable model, where the latent variable is the unobserved confounder. Dorie et al. [Dor+16] extends an Imbens-like approach to accomodate BART as the outcome model. Cinelli and Hazlett [CH20] allow for arbitrary kinds of confounders and propensity score models, but require that the outcome is modeled with linear regression. Cinelli et al. [Cin+19] make concrete assumptions about the edges of a causal DAG and use causal identification tools to translate those assumptions into effect (hence, bias) estimates. However, they assume that all relationships in the DAG are linear. Rosenbaum and Rubin [RR83] assume a categorical covariate and a binary confounder. They don't impose any explicit additional constraints on the propensity score or outcome model, but their general approach requires 4 sensitivity parameters for each level of the observed covariate; making the sensitivity analysis practical requires further assumptions.

A different line of work relaxes parametric assumptions at the price of requiring the analyst to make judgments about more abstract sensitivity parameters [e.g., Fra+19; She+11; VA11; DV15]. For example, Franks et al. [Fra+19] allow arbitrary models to be used for the initial analysis. Their sensitivity model is adapted from the missing data literature, and requires the analyst to specify $P(T = t \mid Y(1-t), X)$—the posterior belief about probability of treatment assignment had the counterfactual outcome under no-treatment been observed. The sensitivity parameters used by these methods are more abstruse than the ones used in parametric-model-based sensitivity analysis. However, the subjective judgments required for each analysis are quite different, and these alternative approaches may be easier in some scenarios. In this sense, these methods are complimentary to the sensitivity analysis approach proposed in this paper.

# 5 Societal Consequences

This paper addressed sensitivity analysis for causal inference. We have extended Imbens' approach to allow the use of arbitrary machine-learning methods for the data modeling. Austen plots provide an entirely post-hoc and blackbox manner of conducting sensitivity analysis. In particular, they make it substantially simpler to perform sensitivity analysis. This is because the initial analysis can be performed without have a sensitivty analysis already in mind, and because producing the sensitivity plots only requires predictions from models that the practitioner has fit anyways.

The ideal positive consequence is that routine use of Austen plots will improve the credibility of machine-learning based causal inferences from observational data. Austen plots allow us to both use state-of-the-art models for the observed part of the data, and to reason coherently about the causal effects of potential unobserved confounders. The availability of such a tool may speed the adoption of machine-learning based causal inference for important real-world applications (where, so far, adoption has been slow).

On the negative side, an accelerated adoption of machine-learning methods into causal practice may be undesirable. This is simply because the standards of evidence and evaluation used in common machine-learning practice do not fully reflect the needs of causal practice. Austen plots partially bridge this gap, but they just one of the elements required to establish credibility.

## Footnotes

[1]See section 4 for a more detailed comparison with related work.

[2]Supplementary material.

[3]Code and data in supplementary material.

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

# Appendix

**Affiliation**
**Address**
`email`

## A   Average Treatment Effect on the Treated

In many situations, the average treatment effect (ATT) on the treated is a more convenient estimand than the ATE.

**Bias**   The same logic we used to derive an expression for the bias of the ATE can be used to derive an expression for the bias of the ATT. For the bias estimand, we just change take the expectation over $X$ in Theorem 3 conditioned on $T = 1$. In practice, the bias can be estimated by taking the mean over only treated units. Note that the reparameterization calculation does not change.

**Calibration using observed data**   Reference values for the ATT can be computed in exactly the same way as for the ATE—i.e., it is not required to restrict the expectations to only the treated units. This is because the bias expression is given in terms of 'full data' $\alpha$ and $R^2_{Y,\mathrm{par}}$.

## B   Proofs

**Theorem 4.** *Under our sensitivity model, Assumption 1, the outcome influence is*

$$R^2_{Y,\mathrm{par}}(\alpha, \delta) = \delta^2 \sum_{t=0}^{1} \frac{\mathbb{E}\big[\psi_1\big(g(X)^t(1-g(X))^{1-t}(1/\alpha - 1) + 1[T = t]\big)\big]}{\mathbb{E}[(Y - Q(T,X))^2]},$$

*where $\psi_1$ is the trigamma function.*

*Proof.*   First, we write:

$$\begin{aligned}
\mathbb{E}(Y - \mathbb{E}[Y \mid T, X, U])^2 &= \mathbb{E}(Y - Q(T,X))^2 \\
&\quad - 2\delta\mathbb{E}[(Y - Q(T,X))(\mathrm{logit}\,\tilde{g}(X,U) - \mathbb{E}[\mathrm{logit}\,\tilde{g}(X,U) \mid X, T])] \\
&\quad + \delta^2\mathbb{E}((\mathrm{logit}\,\tilde{g}(X,U) - \mathbb{E}[\mathrm{logit}\,\tilde{g}(X,U) \mid X, T]))^2 \\
&= \mathbb{E}(Y - Q(T,X))^2 - \delta^2\mathbb{E}[\mathrm{var}(\mathrm{logit}\,\tilde{g}(X,U) \mid X, T)]. \quad \text{(B.1)}
\end{aligned}$$

Where we've used,

$$\begin{aligned}
&\mathbb{E}[(Y - Q(T,X))(\mathrm{logit}\,\tilde{g}(X,U) - \mathbb{E}[\mathrm{logit}\,\tilde{g}(X,U) \mid X, T])] \\
&= \mathbb{E}[\mathbb{E}[(Y - Q(T,X)) \mid T, X, U](\mathrm{logit}\,\tilde{g}(X,U) - \mathbb{E}[\mathrm{logit}\,\tilde{g}(X,U) \mid X, T])]]
\end{aligned}$$

and other standard conditional expectation manipulations.

The usefulness of (B.1) is that $\mathrm{var}(\mathrm{logit}\,\tilde{g}(X,U) \mid X, T)$ has an analytic expression. Namely, by Beta-Bernoulli conjugacy, this is the variance of a logit-transformed Beta distribution. The analytic expression for this variance is,

$$\mathrm{var}(\mathrm{logit}\,\tilde{g}(X,U) \mid X, T) = \psi_1\big(g(X)(1/\alpha - 1) + T\big) + \psi_1\big((1 - g(X))(1/\alpha - 1) + 1 - T\big),$$

where $\psi_1$ is the trigamma function. The claimed result follows by plugging in this expression into (B.1).

$\square$

406  **Theorem 5.** *Under our sensitivity model, Assumption 1,*

$$\alpha = 1 - \frac{\mathbb{E}[\tilde{g}(X,U)(1 - \tilde{g}(X,U))]}{\mathbb{E}[g(X)(1 - g(X))]}.$$

407  *Proof.* The key insight is:

$$\begin{aligned} \text{var}(\tilde{g}) &= \mathbb{E}[\text{var}(\tilde{g} \mid g)] + \text{var}(\mathbb{E}[\tilde{g} \mid g]) \\ &= \mathbb{E}[\alpha g(1 - g)] + \text{var}(g), \end{aligned}$$

408  where the first line is the law of total variance, and the second line uses the assumed $\mathrm{Beta}$ distribution
409  of $\tilde{g} \mid g$. Accordingly,

$$\alpha = \frac{\text{var}(\tilde{g}) - \text{var}(g)}{\mathbb{E}[g(1 - g)]}.$$

410  Now, observe that by the law of total variance,

$$\begin{aligned} \text{var}(T) &= \mathbb{E}[\text{var}(T \mid g)] + \text{var}(\mathbb{E}[T \mid g]) \\ &= \mathbb{E}[g(1 - g)] + \text{var}(g), \end{aligned}$$

411  where we have used that $T \mid g$ is Bernoulli. By the same logic,

$$\text{var}(T) = \mathbb{E}[\tilde{g}(1 - \tilde{g})] + \text{var}(\tilde{g}).$$

412  Whence,

$$\text{var}(\tilde{g}) - \text{var}(g) = \mathbb{E}[g(1 - g)] - \mathbb{E}[\tilde{g}(1 - \tilde{g})].$$

413  The result follows immediately. □