[Reviews · NeurIPS 2020]

Review 1

Summary and Contributions: This paper introduces Austen plots, a method for performing sensitivity analysis of the robustness of causal effect estimates to unobserved confounding. The authors improve upon the method proposed in Imbens (2003) by relaxing some of its strong parametric assumptions. In particular, Austen plots allow the causal modeler to use any model for the propensity scores and expected outcomes. Thus, modelers are free to use flexible machine learning methods to model the observed data--- a practice which is becoming more common. Austen plots inherit other benefits of the Imbens approach: strength of unobserved confounding is assessed in terms of the influence of the confounder on treatment assignment and outcome. This allows the strength of unobserved confounding required to bias the treatment to be compared to the strength of observed confounders. The authors demonstrate the practical value of Austen plots in experiments on the Jobs and IHDP datasets. Additionally, they show that in practice the sensitivity model may be conservative which means robustness claims on the basis of Austen plots may be more trustworthy. ------------------------------------------------------------- Update post-rebuttal: Thanks to the authors for the detailed clarification regarding the arguments by Cinelli and Hazlett. I'm sure the discussion you've added about this point will make the paper better (along with expanded related work as per Reviewer 2's comments). Accordingly, I've raised my score to a 8. I'd like to emphasize that I found this paper a joy to read and want to thank the authors for their excellent work!

Strengths: In general, I thought the paper was strong and well executed. The paper tackles a significant problem: it is important to account for possible unobserved confounding when performing a causal analysis. In describing their method, the authors did a good job of justifying the choices they made in their sensitivity model and explaining how to apply it. I generally agree with the authors claims that the resulting parameterization is easy to interpret. The paper is also well supported by multiple demonstrations of the practicality of the method using examples. I believe that the proposed Austen plots are a generally applicable sensitivity analysis tool that is easy to add to the causal modeler's toolkit.

Weaknesses: One possible criticism is that the work is incremental with respect to the approach in Imbens (2003). The main methodological contribution is the relaxation of parametric assumptions for observed data components. The other benefits, such as the standardization of "the scale of influence strength" and the estimation of the "influence strength of observed covariates for reference" are inherited directly from the Imbens method. I also believe these benefits are shared by other works inspired by the Imbens approach such as Dorie et al. (2016). That said, I think the proposed method is a valuable extension of the Imbens approach: it is easy to use and generally applicable. Practitioners today can add Austen plots to their causal analyses. A more pressing concern I have is regarding the use of observed confounders to judge plausible confounder strengths. Intuitively, this approach is attractive. However, as the authors discuss in the section about "Grouping covariates" (Line 210), collinearity issues make the interpretation of these values difficult. I'm not intimately familiar with the related works, but I believe there is a [strong] criticism of this general methodology in, e.g., Franks et al. (2019) and Cinelli & Hazlett (2020). Cinelli & Hazlett state "This practice [of benchmarking against observed covariate statistics] has undesirable properties, because the observed statistics used as benchmarks are themselves affected by the omission of the confounder." I am hoping the authors can address the criticisms levied by these works, and how they apply to the proposed methodology?

Correctness: To the best of my knowledge the paper is technically sound. The demonstrations of the method were also well done.

Clarity: This paper was very enjoyable to read. The authors did a great job motivating and overviewing their method in the introduction, and I appreciated the incorporation of examples throughout the text. The explanation and derivation of the sensitivity model was also easy to follow. Given more space in the revision I think the paper would benefit from a brief discussion or conclusion to close the paper. Additionally, I think it would be helpful for the authors to explain up front what is expected to happen on the four Jobs dataset variants when they discuss the dataset around Line 244. In particular, originally I was unsure of what the purpose of the "restricted-sample" dataset was and how results would differ relative to the other observational variants.

Relation to Prior Work: As mentioned above, I'm not intimately familiar with the related work but I believe the authors did a good job of clarifying how the proposed method differs from existing methods. As I mentioned in the weaknesses, I do think it is important for the authors to discuss and directly address the concerns raised in, e.g., Cinelli & Hazlett (2020).

Reproducibility: Yes

Additional Feedback: I think the scales on the plots in the paper (i.e., x and y axis limits in [0, 1]) cause the plots to have a lot of uninformative white space, since generally there aren't points high above the sensitivity curves. I think the plots would be visually improved by truncating the axes. This way it is easier to distinguish where the individual covariates fall on the plot.


Review 2

Summary and Contributions: This paper proposes a nonparametric generalization of a sensitivity analysis model from Imbens (2003), for use in causal inference when unmeasured confounding assumptions are violated. An expression for the bias of the usual adjusted population ATE is given, along with an alternative R^2-style parametrization of one of the sensitivity parameters. Plausible values of the sensitivity parameters are obtained by estimating analogous quantities as if one or more of the observed covariates acted as the unmeasured confounder, and results are displayed in so-called Austen plots. The approach is illustrated in several real examples. ----- Update post-rebuttal: I found the authors' response to be thorough and thoughtful. I continue to feel this paper will make a useful and unique contribution.

Strengths: All claims appear sound, both theoretically and empirically. The work gives a novel and creative but also practically useful generalization of a popular sensitivity model. The work is solidly grounded in flexible nonparametric modeling, while also providing easy-to-interpret calibration measures and graphical displays. This paper is unique in making important advances both theoretically and practically. I expect it will be very relevant to the NeurIPS community.

Weaknesses: To me the main weaknesses are a lack of attention to issues of inference, and a somewhat underwhelming discussion of how the developments compare to previous work in sensitivity analysis and elsewhere. These seem relatively easy to remedy. Regarding inference and statistical uncertainty, it would be nice if the authors gave a little more detail. The bias in Theorem 3, the reparametrized quantity in Theorem 4, and the leave-one-out R^2 measures in Section 2 are all unknown quantities depending on P that need to be estimated. Simple plug-in estimators are proposed, but those will all likely be suboptimal in realistic nonparametric settings. It might be worth mentioning other approaches for estimating these quantities, as well any approaches for inference/uncertainty quantification. For example, one could imagine Austen plots with measures of uncertainty attached, e.g., confidence balls along with point estimates. Regarding relation to previous work, see comment below.

Correctness: The claims and empirical methodology appear correct.

Clarity: This paper is extremely well-written and clear.

Relation to Prior Work: More examples of relevant work could have been cited and compared. There is a quite long history of sensitivity analysis in causal inference, but the Related Work section focused on a relatively small segment. One prominent omitted paper is Robins, Rotnitzky, & Scharfstein (2000), which proposes extensive non- and semi-parametric methods in a wide array of general sensitivity analysis problems, and appears to be the first to emphasize the importance of using models that do not restrict the observed data, also central to the authors’ work. Particular examples of their general approach are described in Brumback et al (2004) and elsewhere. A more thorough comparison against this and the many papers since would be welcomed. A few other omitted references include the leave-one-covariate-out method of Lei et al. (2018) and the variable importance approach of Williamson et al. (2020+), both of which study quantities similar to the leave-one-covariate-out R^2 measure used by the authors. Other related work could also be found in the references of these papers.

Reproducibility: Yes

Additional Feedback: 1) I do not really see why alpha is easier to interpret than delta. Both are coefficients in regression models (on X and U), and in fact the latter is a coefficient in a linear model, which seems like it could even ease interpretation? Perhaps the authors mean that delta is a coefficient for a variable that is an unusual transformation of U and X (though see comment 2 below)? Would we really expect to have more knowledge of this than for a generic unmeasured confounder whose exact scale might typically be unknown? 2) Are the delta parameters in Eq 3 and Assumption 1 the same? It seems there is a scaling by alpha that makes them different, which might be worth pointing out and/or distinguishing notationally. 3) An alternative approach to specifying sensitivity parameters exactly and varying them is to use bounds. Do the authors have any ideas about how such an approach could be combined with their proposal? For example, one might only be willing to assume the values of alpha/delta are within some range, rather than taking particular specific exact values. 4) I think the authors’ claim that other nonparametric sensitivity parameters (described in the 2nd paragraph of the Related Work section) are more “abstract” and “abstruse” is too subjective, and needs to be changed or else formalized. In my view these parameters are less abstract because they are more fundamental, not being tied to particular & likely misspecified parametric models. 5) Related to the points above and earlier about relations to previous work, I think it would be helpful if the authors gave more detail comparing and contrasting their sensitivity parameters to those appearing before. This would help readers judge which may be most useful in their particular setup, as well as help distinguish how the paper relates to previous work.


Review 3

Summary and Contributions: The paper addresses sensitivity analysis in causal inference, specifically, the bias of the estimated ATE relative to the strength of unobserved confounding. Its main contribution is developing a sensitivity model to formalize the strength of the unobserved confounding on treatment and outcome with interpretable sensitivity parameters that allows for arbitrary propensity score and conditional outcome models for observed data.

Strengths: The proposed sensitivity analysis method is novel and should be useful in practice.

Weaknesses: I don’t see any specific issues with the paper per se. How significant the results of the paper are may be arguable.

Correctness: The claims look like correct.

Clarity: The paper is well written.

Relation to Prior Work: The relation to prior work is clearly discussed.

Reproducibility: Yes

Additional Feedback: The paper mentions the proposed method is “tractable” and motivated by tractability while “The logit linear model does not directly lead to a tractable sensitivity analysis”. Can you elaborate on this point? ===After author rebuttal === My opinion has not changed. The paper makes enough new contributions to be accepted.

[Author Response · NeurIPS 2020]

We thank the reviewers for the insightful comments and questions, and for their support. Using the camera ready 9th page, we have added a discussion section, and expanded the related work.

In the new discussion, we emphasize that Austen plots are most useful in situations where the conclusion from the plot would be 'obvious' to a domain expert. For instance, the LaLonde RCT plot shows that a confounder would have to be much stronger than the observed covariates to induce substantial bias. Similarly, the LaLonde observational plot shows that confounders similar to the observed covariates could induce substantial bias. Such conclusions would not be affected by mild perturbations of the dots or the line. By contrast, Austen plots should not be used to draw conclusions such as, "I think a latent confounder could only be 90% as strong as 'age', so there is a small non-zero effect". Such nuanced conclusions might depend on the particular sensitivity model we use, or statistical misestimation or incautious interpretation of the calibration dots—the latter two concerns raised by the reviewers. Drawing precise quantitative (rather than qualitative) conclusions about induced bias from Austen plots would require careful consideration of these issues, and expert statistical guidance. Hence, we recommend that the plots should be used mainly with domain experts to guide qualitative conclusions ("this job program likely works", "this study doesn't establish drug efficacy").

**R1** *Cinelli and Hazlet* (CH) warn against 'informal' benchmarking procedures. Our bias calculation is based on a formal bound, so this critique doesn't directly apply. To illustrate the issue, CH provide an example where the change in ATE induced by leaving out an observed covariate $X$ is smaller than the bias induced by omitted variable $U$, even though $X$ and $U$ come into the model in an identical manner. In contrast, the bias estimate in the Austen plot model is typically higher than the change in ATE induced by leaving out a variable—this is shown in our model conservativism experiments. In particular, Austen plots (correctly) anticipate that the bias from omitting $U$ can be higher than the ATE difference induced by computing without $X$, even when $U$ and $X$ have identical confounding strength.

The CH example draws attention to a point that requires some care. The crux of it is 1. strength of influence of $U$ is computed *conditional on the observed covariates*, and 2. the reduction in uncertainity in going from only $X$ to both $X, U$ may be greater than the reduction in uncertainity in going from nothing to $X$. Thus, it's possible in principle for the influence of $U$ to be larger than the estimated (observed) influence of $X$, even if $X$ and $U$ are similar causal variables. In other words, the omission of $U$ could cause us to underestimate the influence of $X$ from the observed data. Accordingly, when the domain expert compares the strength of the unobserved confounder to a reference dot for $X$ on the plot, they must also ask if knowing $U$ could have substantially increased the predictive power of $X$. In cases where this seems plausible—e.g., the domain expert expects an important interaction between $U$ and $X$—then naively eyeballing the dot vs line position may be unreliable and further careful thought is required. However, we note that examples of this kind are somewhat contrived. Indeed, we usually expect the opposite effect. If $U$ and $X$ are dependent, then some of the information in $X$ will be redundant, and the measured $R^2$ and $\alpha$ will *overestimate* the true influence. That is, the CH effect tends to make our sensitivity analysis conservative. This is reflected by the fact that grouping similar covariates (to mitigate redundancy) led to higher computed influence in every example considered in the paper. Although the CH effect is an important conceptual point and the domain expert should consider it as part of due diligence, it doesn't seem to have much impact on the practical use of Austen plots. We also note that this conceptual subtlety is a generic feature of calibrating sensitivity analyses, not particular to our method. This is reflected, e.g., in Franks et al §5.2.1, which describes their calibration procedure based on looking at variance explained by observed $Z$ conditioned on $X \setminus Z$—a procedure similar to ours, carrying the same nuance. We have clarified this point in the newly added discussion section. We thank the reviewer for bringing this to our attention.

**R2** *Inference* We agree that inference is a key issue. We have added some additional detail estimation via the plug-in estimators. For handling uncertainty, we suggest bootstrapping several plot versions and trusting only conclusions supported by all plots. We tried this on the examples in the paper, and found no change in conclusions. We have added an appendix describing how to visually summarize the uncertaintity; see fig (cf. Fig 1 in paper). Note: $\alpha$ uncertainty is plotted, but too small to show up clearly. Formal uncertainty quantification is an interesting direction for future work; particularly important because bootstrapping requires model refitting, which can be computationally intensive for the ML models that motivate the paper.

We emphasize that the unobserved confounder bias we address exists even in the (very) large data regime where plug-in estimators work very well. Indeed, this is the setting where misidentification bias matters most relative to statistical error. Accordingly, we believe the paper, using the plug-in estimators, is a significant contribution even in the absence of efficiency guarantees. In the discussion, we caution that the plots may be misleading if there's large uncertainty in the estimaton of $Q$ and $g$. As part of the discussion on estimation issues, we note efficient estimation as a good direction for future work.

*Related work* Thank you for the pointers to the additional related literature. As you suggest, we have used the extra page to substantially expand the related work section.

[Meta-Review · NeurIPS 2020]

The paper makes an important contribution to the important problem of sensitivity analysis. The paper is very well written and presented, and the technical contribution is interesting and would surely prove useful for practitioners of modern causal inference methods. The reviewers made many useful suggestion which I trust the authors will incorporate into their work. I also suggest citing [1], which focuses on the issue of calibrating the sensitivity analysis parameters to the observed confounders. [1] Hsu, Jesse Y., and Dylan S. Small. "Calibrating sensitivity analyses to observed covariates in observational studies." Biometrics 69.4 (2013): 803-811.